# A Hybrid Ontology-Based Recommendation System in e-Commerce

**Márcio Guia [1,\*], Rodrigo Rocha Silva [2,3,\*]** 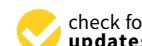 **and Jorge Bernardino [1,3,\*]**

[1] Coimbra Polytechnic –Instituto Superior de Engenharia de Coimbra (ISEC), 3030-190 Coimbra, Portugal
[2] FATEC Mogi das Cruzes, São Paulo Technological College, Mogi das Cruzes 08773-600, Brazil
[3] Centre of Informatics and Systems of University of Coimbra (CISUC), 3030-290 Coimbra, Portugal
[\*] Correspondence: marcioguia13@hotmail.com (M.G.); rrochas@dei.uc.pt or rrochas@gmail.com (R.R.S.); jorge@isec.pt (J.B.)

**Abstract:** The growth of the Internet has increased the amount of data and information available to any person at any time. Recommendation Systems help users find the items that meet their preferences, among the large number of items available. Techniques such as collaborative filtering and content-based recommenders have played an important role in the implementation of recommendation systems. In the last few years, other techniques, such as, ontology-based recommenders, have gained significance when reffering better active user recommendations; however, building an ontology-based recommender is an expensive process, which requires considerable skills in Knowledge Engineering. This paper presents a new hybrid approach that combines the simplicity of collaborative filtering with the efficiency of the ontology-based recommenders. The experimental evaluation demonstrates that the proposed approach presents higher quality recommendations when compared to collaborative filtering. The main improvement is verified on the results regarding the products, which, in spite of belonging to unknown categories to the users, still match their preferences and become recommended.

**Keywords:** recommendation system; ontology; collaborative filtering; KNN; data mining

## 1. Introduction

Over the last years, due to the growth of the Internet, the amount of data found in web pages and social networks has increased significantly. Hence, although users have more information available to them, it becomes more and more difficult to meet their demands when it comes to oferring information related to their preferences [1]. The growth of the Internet has also potentiated the proliferation of e-services over numerous online platforms, whose main advantage is to provide products and services anywhere and anytime to users who have not yet purchased them [2]. With this amount of data and services available, it is not only difficult for users to quickly find items in which they are interested in, but also for e-commerce and similar systems to recommend items among the available data. In order to resolve these problems, recommendation systems have been offered as a solution.

One of the most common successful methods in recommendation systems is collaborative filtering [3], which consists of recommending products and items that have similar preferences among users to the active user who liked or purchased them in the past.

Another common recommendation system method is content-based [4], which consists of recommending products or items with similar characteristics to those already purchased by the active user. The main problem of this method is that the products recommended by the system will probably be very similar to those the active user has already purchased.

Another common group of recommenders is knowledge-based recommenders, which consists of recommending items to the active user according to the domain knowledge about items that meet the

user's preferences [4]. One type of knowledge-based recommenders is ontology-based, which can be defined as an explicit specification of a conceptualization [5]. The ontology-based recommenders are knowledge-based recommenders that use an ontology to represent knowledge about the user and items [4]. The ontology represents knowledge about the user, the products, and the relationship between them.

Current research demonstrates that the ontology-based method improves recommendation systems by resolving the most common limitations of traditional systems. However, building an ontology-based recommendation system is a complex and time-consuming task, which demands knowledge engineering skills [4,6].

In this paper, we propose an innovative recommendation system approach, which combines the simplicity of the most prevalent algorithm in collaborative filtering, which is the K-Nearest Neighbor algorithm (KNN) [4] with the efficiency of ontology-based recommenders.

Fundamentally, we create four main classes: Person, User, Neighbour, and Product. These classes are used to represent and model knowledge and details about each active user and the relationship between them. In order to recommend items to the active user, firstly we apply the KNN algorithm to find the nearest neighbors, and then we reapply it to find the nearest Products to recommend to the active user.

The main contributions of this work are the following:

- We propose a new e-commerce recommendation system, which combines collaborative filtering with ontology-based recommenders;
- The proposed recommendation system considers not only the users with similar preferences to the active user but also obtains knowledge about the user, their neighbours, products and the relationship between them;
- Our proposal increases the number of recommended products from categories which the active user has not yet purchased;
- The proposed system is scalable, which means that it maintains good level performance when the workload increases.

The rest of this paper is organized as follows. Section 2 presents some related work, focused on ontology-based and collaborative filtering recommenders. Section 3 describes the hybrid ontology-based approach, namely, the ontology classes and their properties, and the application of the KNN to find the K nearest Neighbours and Products. Section 4 describes the experimental evaluation and discussion of the results. Finally, Section 5 concludes the paper and presents future work.

## 2. Related Work

In this section, we present a summary of the research papers that focus on ontology-based and collaborative filtering recommenders, in different areas, such as e-commerce, e-learning, and social events.

Obeid et al. [7], propose a hybrid recommendation system, which combines an ontology-based recommender with machine learning techniques. The main goal of the proposed approach is to recommend universities to the students using an ontology to represent domain knowledge about the universities and students, and machine learning techniques to perform the recommendation. The proposed approach not only focuses on the students' grades, but also on their skills and interests, which is an innovative idea. However, it is necessary to perform an experimental evaluation to confirm the effectiveness of the hybrid recommendation system.

Ayundhita et al. [8], suggest an ontology-based recommender that aims to recommend laptops to users who are not aware of low-level specifications. The proposed approach uses the ontology to map functional requirements with low-level specifications. Basically, the system asks functional requirements to the user according to their preferences and then the ontology maps those requirements with low-level specifications to find laptops that match the active user's preferences. Although it

is an innovative idea, this approach requires an effort of the active user, who needs to introduce the functional requirements of the intended laptop. Therefore, when the active user does not know enough functional requirements, the recommendation system does not work. The experimental evaluation shows that the proposed approach achieves an accuracy of 84,6% when compared to general recommender systems that achieve a low accuracy of 61.5%. However, the experimental evaluation was performed with only 39 users, which is not a large enough number to conclude the improvement of accuracy.

Zehra et al. [9], propose an approach that uses an ontology-based recommender to provide schools that matches the active user's preferences. This work uses the active user comments on their Facebook feed about a certain school to extract features that allow the creation of an ontology-based recommender, based on the polarity of the comments. Moreover, the recommendation process can recommend schools based on certain branches chosen by the active user. Despite the differences between this work and our proposal, this paper allows to perceive that ontology-based recommender can improve the quality of the recommendation process.

Cutolo et al. [2], present an approach that combines the use of two ontologies, in order to retrieve e-services' descriptions, such as the products' commercial features, and creates a user profile that is composed of transactional data, user personality, habits, and preferences. The aim of the proposed approach is to recommend products to users that match the user profile information together with the products and services description. Although the proposed approach seems to be original, there is no experimental evaluation to confirm the proposed idea.

An approach to classify events is presented in [10], which uses an ontology to define the most important properties in describing an event, based on the LODE ontology [11]. Instead of classifying an event into categories, this work aims to classify an event according to tags. For experimental evaluation, a synthetic dataset based is generated on real data from a popular social network. The results when compared to the LODE ontology, show an improvement of correctly classified events, a decrease of memory consumption and better time performance. Although the proposed idea achieved good results and proved the advantages of the usage of the ontology, experimental evaluation with real data is not presented.

Feng et al. [3], propose an approach to improve the collaborative filtering recommender. The main goal is to improve the accuracy and quality of the recommendation system, mainly when the sparsity of data is a problem. The proposed approach consists in the use of three impact factors in order to find the similarity between users. The impact factors that are included in the similarity calculation between users are not only the co-rated items but also all data available between the two users. The pairs of users with low co-rated items will be punished, and the third impact factor considers the user rating preferences. However, the third impact factor, in our opinion does not reflect the preferences of the active user, because if a user has only rated two products and has had a bad experience with them, it does not mean that s/he only searches for products with low rating. The experimental evaluation shows that the proposed approach not only resolves the sparsity problem but is also more suitable for recommendation systems with this issue. For that reason, this approach does not improve accuracy in datasets that do not suffer from sparsity and only focus on the ratings given to the products, ignoring the preferences and characteristics of the active user.

An analysis of ontology-based recommenders research papers for e-learning services from 2005 to 2014 is proposed in [4]. The authors conclude that ontology-based recommender papers for e-learning have increased over the years and can provide better results in the recommendation process compared to traditional recommendation techniques, such as collaborative filtering and content-based. Another conclusion is that, in the future, one of the trends of recommendation processes is hybridization of ontology-based recommenders with other known techniques, in order to improve the accuracy and performance of the recommendation process. A similar work is provided in [6] that presents an analysis of ontology-based recommenders research papers for e-learning services from 2010 to 2018, which mainly shows the research papers that only use ontology and the ones which combine

hybrid techniques with the use of ontology. The authors conclude that although the ontology-based recommender approach to e-learning services achieves better results when compared to conventional recommendation techniques, the ontology-based approach has some limitations, which is the example of being time-consuming and requiring knowledge engineering skills. They also conclude that the hybridization of conventional techniques with ontology-based recommenders for the e-learning domain can achieve better results when compared to any other technique.

The authors of [12] present an hybrid system, EntreeC, that combines knowledge-based and collaborative filtering to recommend new restaurants to the active user. To apply knowledge-based, the active user introduces a restaurant that s/he likes or the ideal characteristics that s/he expects among the characteristics available on the system, and then the system finds similar restaurants. As restaurants are being recommended by the system, the active user can accept the proposed restaurant, browse to others or evaluate the proposed one, which means that the process of recommendation requires an effort of the active user and can be time-consuming. The system creates a user profile according to his/her navigation, which allows the application of the collaborative filtering to choose similar users and then recommend the restaurants that match the active user preferences. However, collaborative filtering approaches compare only users whose reactions are largely identical, which means that predictions will be based on a much smaller number of users and hence the process of recommendation using collaborative filtering perhaps does not improve the knowledge-based step. Furthermore, if a user exits when the system recommends a new restaurant this is interpreted as the user likes that restaurant. This can be true but, in our opinion, this information cannot be used as a positive because it can also mean that the user gives up searching for restaurants.

An extensive study of research between 2005 and 2015, only takes into account hybrid approaches to deal with recommendation systems is provided in [13]. After a selection process, the authors choose only the best 76 papers according to 7 main questions. According to the authors, among the 76 papers, only 3 focus on providing diversity in the recommendations and only 11 focus on scalability, which is precisely the two main goals of our work. The authors conclude that one of the trends in the future is combining knowledge-based recommender with collaborative filtering, which is what we present in our work.

Jiang et al. [14] recommend an improvement of the Slop One algorithm, solving the problem of low prediction accuracy and poor performance when dealing with personalized recommendation tasks that concern the relationship among users. In order to solve these problems, it is suggested that an improvement in Slop One algorithm be made based on trustworthy data and user similarity. Although the proposal solves the common problems of the basic Slop One algorithm, the use of trustworthy data and user similarity only takes into concern to the ratings given to the products, which means that the recommendation process does not include the preferences of the active user, neither the features of the products.

The development of a utility-based recommender system to deal with clothing recommendations is presented in [15]. This work uses the ratings given by the users to the products implicitly, which avoids user effort. Although the process of recommendation seems to be original, it will recommend products very similar to the ones that the active user already purchased, which our work looks to avoid. Moreover, in e-commerce platforms that deal with a huge number of products and where these differ from the number of attributes and features, the proposed approach does not work.

Jyoti et al. [16] propose a collaborative filtering approach to recommend products based on active user' ratings. The proposed approach is the use of basic collaborative filtering without any improvement. After selecting the top products to recommend to the active user, the proposed approach finds the nearby store, which has the recommended product. The proposed approach only works in e-commerce platforms that have the location of the products and the location of the active user. This work differs from ours on the basis that it only focuses on the ratings given to the products, hence, it does not include the preferences of the active user, neither the features of the products.

A clothing recommendation system based on the improvement of user-based collaborative filtering algorithm is defined in [17]. In order to improve the basic user-based collaborative filtering algorithm, the authors define a better calculus of similarity among users and also a user-item linked list to reduce sparseness of the matrix. This work gives more importance to the products that are unpopular, but it does not specify the percentage of products in the chosen dataset, which raises doubts about the overall quality of these products. Furthermore, the use of a linked list can perform well on small datasets, however, when dealing with large amounts of data, this approach will increase the time of the recommendation process since the entire linked list will have to be completely scrolled. This work also only focuses on ratings given to the products, which means that the recommendation process does not include the preferences of the active user, neither the features of the products.

Bozanta and Kutlu [18], describe a hybrid recommendation system which combines collaborative filtering and content-based filtering together with contextual information like the weather or the season of the year. The experimental evaluation shows that this hybrid approach can improve the quality of recommendation systems over the traditional approaches. The main difference between this work and ours is that the first one requires an unnecessarily large amount of complexity to build the proposed approach since it uses three recommendation techniques along with a neural network.

Nilashi et al. [19], present a hybrid recommendation system which combines collaborative filtering with ontology and dimensionality reduction techniques in order to improve the sparsity and time complexity of the collaborative filtering approach. The experimental evaluation shows the benefits of building a hybrid recommendation system when compared to the traditional systems. Apart from the similarities to our work, this approach focuses more on the ratings given to movies. In addition, it does not give attention to the diversity of the movie's recommendation, which is one of our main contributions.

All the previous papers are focused on improving the quality of the recommendation process, but the ones that propose approaches based on knowledge-based requires effort from the active user, which is what we look to avoid with our approach. Those who try to improve collaborative filtering approaches only take into concern the ratings given to the products by the users, which means that they do not include the knowledge about the active user in the recommendation process, active user's neighbours, products nor relationships between them. Our work presents a different approach that combines knowledge-based recommenders with a collaborative filtering approach, which is translated into recommendations with more diversity. Furthermore, we also give special attention to the scalability of the proposed approach. One of our main goals is to provide recommendations of products that not only receive high scores from the respective neighbours but also from all users that have purchased them.

## 3. The Hybrid Ontology-based Approach

This section presents the approach used to build the recommendation system and as previously mentioned, combines the application of ontology and the KNN algorithm. Figure 1 shows the overview of our hybrid ontology-based approach, which starts with creating a user profile that contains, among other attributes, the categories and respective products bought by the user.

In order to recommend products to the user, it is necessary to find other users (Neighbours) that at least have bought one common product, and select only the ones that, respecting the previous criteria, have also bought other products. These are the products that have the potential to be recommended to the user.

When the user and their Neighbours' profiles are created, we apply the KNN algorithm to find the nearest Neighbours of the user. This procedure allows the extraction of products that the user has not yet purchased. The last step, is to apply the KNN algorithm to find the nearest products according to our metrics, which include the overall rating given by the respective neighbor, the price of the product for knowing wheter it is between the lowest and highest value spent by the active user. We also include a text review of each product because it is a huge source to extract important information [20].

In the following subsections, we provide a detailed explanation about each step of the proposed approach. This includes the dataset used, the ontology described, and the application of the KNN algorithm.

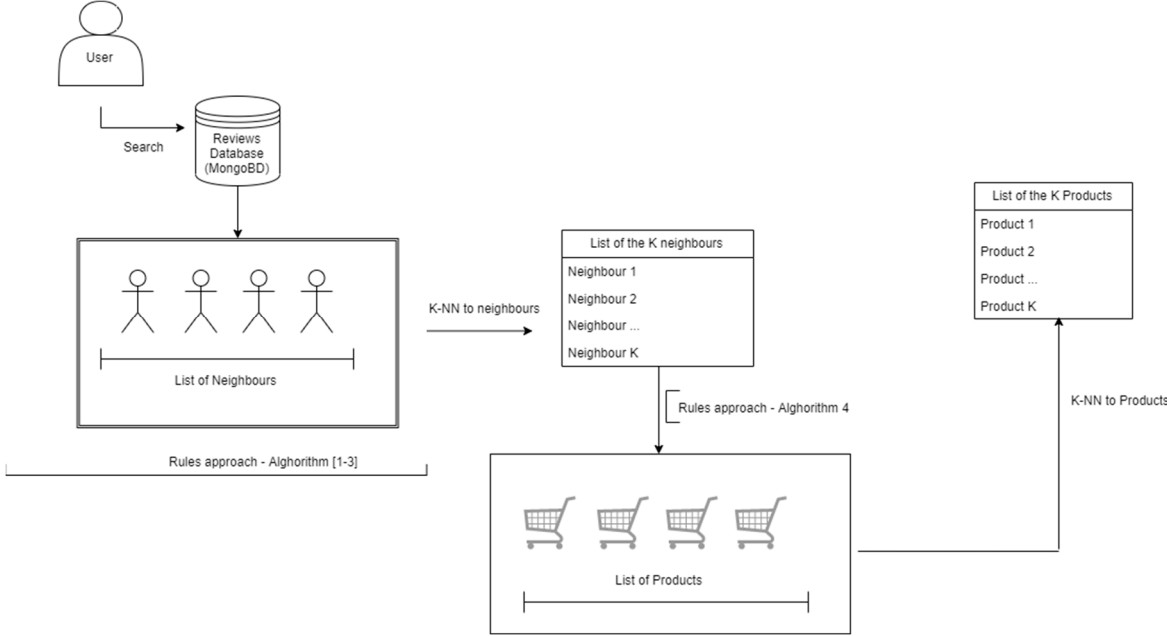

**Figure 1.** Overview of the hybrid ontology-based approach.

### 3.1. Dataset

The dataset used to build the recommendation system consists of 142.8 million reviews from Amazon between May 1996 and July 2014 [21]. The authors of the dataset provide a small version of this dataset, which we used to build our recommendation system. The smaller version of the dataset consists of 82.63 million reviews, which is then divided into two parts. The first one takes into account revisions and has attributes such as reviewerID (string), productID (string), reviewerName (string), reviewerText (string), overall (numeric) and reviewTime (string). The second part concerns itself with metadata, which includes a description of the products, such as productID (string), title (string), price (numeric), related products that Neighbours also bought and/or viewed, bought_together and/or bought after viewing. Metadata also includes brand (string) and categories (list of categories that product belongs).

From this dataset, we create the reviews database (MongoDB), which contains the first 300,000 reviews of 16 categories, selected at random. We choose the MongoDB database to save the data because it is one of the most popular NoSQL databases [22]. Moreover, the authors of [23] mention that it can achieve good performance when referring to reading operations, which is the main operation used in our approach.

### 3.2. The ontology model

In order to improve the Product recommendation to the active user, we combine, as previously mentioned, an ontology-based recommender with a collaborative filtering approach. According to the guide proposed in [24], that is used to create the ontology model, an ontology can be defined as a formal explicit description of concepts in a domain.

This formal explicit description includes the classes that describe the concepts of the respective domain. In order to describe the features and attributes of the concepts properties used, each property has their own facet, which is the property's type.

In the following two subsections, we explain the domain and scope of the proposed ontology model and also the respective classes and their properties.

3.2.1. The domain and scope of the ontology model

The main purpose of creating and using an ontology in the recommendation process is to implement a data model that can describe and represent all the concepts that concern to the active user, his neighbours and the products purchased by the latter, which are the ones that can be recommended to the active user. Therefore, one of the main applications of this ontology will be e-commerce platforms.

In order to define the domain and scope of the ontology model, we create a list of questions that help find if the ontology model contains enough information to describe the domain of the recommendation process. The list of the defined question is divided into two categories, namely the questions that concern the active user and the respective neighbours and also the questions that concern the products purchased by the neighbours.

The questions made to the active user and his neighbours are the following:

- Who is the active user?
- Which products and respective categories has the active user purchased?
- What was the average amount spent by the active user as well as the maximum and minimum amount?
- Who are the neighbours of the active user?
- What is the relationship between the active user and each one of the neighbours?
- Who are the neighbours that have made purchases in different product categories as those of the active user?

The second part of the questions, which concern to the products purchased by the neighbours are the following:

- Can the product match the active user's preferences?
- What is the rating given by active users's the respective neighbour?
- Which are the five most common words used in all the revisions of the product?

3.2.2. Ontology model classes

In order to develop the ontology-based recommendation system, four main classes are created. The four classes of our ontology model are Person, User, Neighbour, and Product. In order to create them, we have followed the top-down method [25], which consists of starting the development of the class hierarchy by the most general concepts of the domain and subsequent specialization of the concepts.

Figure 2 shows the relationship between the four classes and their respective attributes.

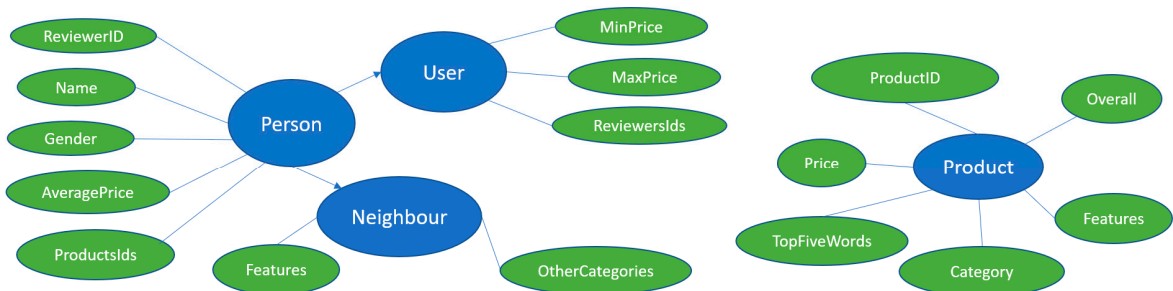

**Figure 2.** Ontology used, respectively classes and attributes.

The classes and attributes are the following:

- **Person:** a class that represents a person that have made reviews of products, with the following attributes:

    o　　ReviewerID—identifies the person who made the review;

- o    Name—name of the person;
- o    Gender—identifies the gender of the reviewer based on the Name;
- o    AveragePrice—the average amount spent by the reviewer;
- o    ProductsIds—represents the Ids of products purchased by the reviewer.

- **User:** a subclass of person that represents the user to whom the system will recommend new products. This class has the same attributes as Person plus the following:

  - o    MinPrice—the price of the product with the lowest value of all products purchased by the user;
  - o    MaxPrice—the price of the product with the highest value of all products purchased by the user;
  - o    ReviewersIds—identifies the IDs of reviewers that have purchased at least one product from the ProductsIds list.

- **Neighbour:** a subclass of person that represents a reviewer that purchased at least one product from the ProductsIds of User and purchased different products. This class has the same attributes as Person plus the following:

  - o    OtherCategories—represents whether a Neighbour has purchased products in different categories of User;
  - o    Features—represents the features vector that will be used to calculate the Euclidean distance and find the nearest Neighbours of User.

- **Product:** a class that represents a product purchased by a Neighbour. This class has the following attributes:

  - o    ProductID—identifies the product;
  - o    Price—the price of the product;
  - o    Overall—identifies the score given by the Neighbour in the revision and can be a value between 1 and 5;
  - o    Category—the category of the product;
  - o    TopFiveWords—identifies the 5 most common words in all the reviews of the product, concerning all persons who purchased the product;
  - o    Features—represents the features vector that will be used to calculate the Euclidean distance and find the nearest products to recommend to User.

Table 1 shows the classes Person, User, Neighbour, Product and the corresponding properties and facets of the ontology.

The next subsections explain how the recommendation system works. This includes the process of creating the classes and applying the KNN algorithm, finding the nearest Neighbours and then the nearest products according to the metrics.

**Table 1.** Classes, properties and facets of ontology.

| Class | Properties | Facets |
|---|---|---|
| Person | ReviewerID | string |
| | Name | string |
| | Gender | number |
| | AveragePrice | number |
| | ProductsIds | list |

**Table 1.** *Cont.*

| Class | Properties | Facets |
|---|---|---|
| User | MinPrice | number |
| | MaxPrice | number |
| | ReviewersIds | list |
| Neighbour | OtherCategories | boolean |
| | features | vector |
| Product | ProductID | string |
| | Price | number |
| | TopFiveWords | list |
| | Overall | number |
| | Category | string |
| | Features | vector |

*3.3. Rules Approach—Algorithm 1, 2 and 3*

In this subsection, we detail the first step of the recommendation system, which starts by creating a user profile. Algorithm 1 presents the first step, which is the rule to extract the products purchased by the user.

---

**Algorithm 1:** Find reviews made by the active user

---

**Input:** ID of the active user (string)
**Result:** List of reviews that active user have made
**return** reviews on reviews Database that have revision.reviewerID equal to ID of the active user

---

The first step consists of searching the database for reviews that have the same reviewerID as the active user. After extracting the revisions made by the user, we create a User Profile, with the attributes explained in Section 3.2.

Algorithm 2 shows the process to find the IDs of all reviewers that have purchased at least one product from the ProductsIds from the User Profile.

---

**Algorithm 2:** Find IDs of reviewers that have bought at least one product of the List of products from active user

---

**Input:** List of IDs product that active user have purchased
**Result:** List of distinct elements which contains the IDS of neighbours who have made at least one review that have revision.ProductID equal to the ID of the active user
**return** List of IDs of neighbours

---

Observing the pseudocode of Algorithm 2, we see that for each element in the list of products IDs from the active user, a search in Reviews Database is made, in order to find the IDs of all the reviewers. This excludes the active user, who bought the same product. Then these IDs are stored in the global variable idsNeighbours. After completing this procedure to all elements in the list of products Ids from the active user, we select the distinct elements in idsNeighbours.

In order to avoid applying the KNN algorithm to Neighbours who didn't buy different products from the user, it is fundamental to select only the reviewers that have purchased other products. This procedure is explained in Algorithm 3.

---

**Algorithm 3:** Find if Neighbour has bought different products from the active user

---

**Input:** ID of Neighbour, list of products of the active user
**Result:** The condition about wheter Neighbour has made at one least review which not have review.ProductID in the list of products of the active user
**return** True or False

---

From the pseudocode presented in Algorithm 3, we see that, in order to find whether a Neighbour has purchased different products from those the active user has, it is necessary to access the Reviews Database and search for the products that the first has purchased and then check each of them in order to find out whether they belong or not, to the active users's list of products. At the momement, the previous condition is found to be true, the procedure stops and creates the respective Neighbour Profile, as previously explained in Section 3.2. The previous procedure is performed to each element of the attribute ReviewersIds of active User.

*3.4. KNN Algorithm to Find Neighbours*

As mentioned before, the KNN algorithm is used to find the nearest Neighbours among the Neighbours of the user. In order to apply the Euclidean distance, we need to create a vector of features for each Neighbour. Figure 3 shows an example of this procedure.

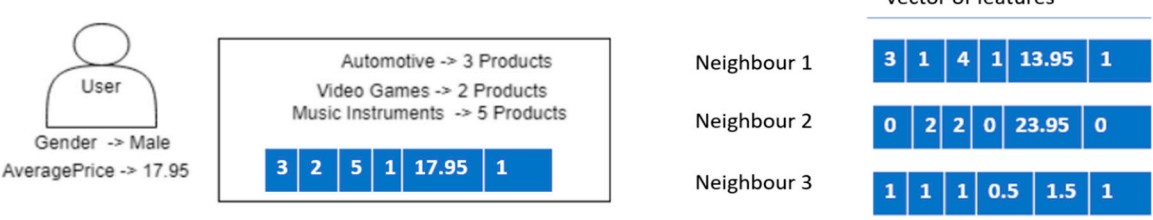

**Figure 3.** Creation of the vector of features to each Neighbour.

Figure 3 shows that, the active user's gender is male, has spent an average of $17.95 and has purchased products in three distinct categories, namely 3 Products in Automotive, 2 Products in Video Games and 5 Products in Music Instruments. The first 3 columns in the Neighbours vector of features concerns the number of products that each neighbor has purchased in each category. The fourth column identifies the gender (1 to male, 0 to female and 0.5 when it is not possible to discover gender). The fifth column is the average money spent, and the last identifies if a Neighbour has purchased products in distinct categories from the user, meaning 1 if true and 0 if false. Our goal with the last column is to give priority to the Neighbours who have purchased in other categories. After creating all the Neighbours vector features and normalizing the columns in a range of 0 and 1, we calculate the Euclidean distance between each one of them and the user vector feature, to discover the nearest Neighbours of the user.

*3.5. Rules Approach—Algorithm 4*

When the nearest Neighbours are defined, as explained in the previous subsection, it is necessary to find out whether the IDs of the products that they have purchased are new to the active user. This procedure is explained in Algorithm 4.

---

**Algorithm 4:** Find IDs of products bought by neighbours that active user don´t bought

---

**Input:** List of all Products bought by the active user´s neighbours
**Result:** List of ProductsIds that active user doesn't have yet bought
**return IDsProducts**

---

In Algorithm 4, we see that for each element in the list of all ProductsIds bought by Neighbours, the procedure consists of making sure that the product does not belong to the attribute ProductsIds of the active User. Only those that meet the above condition are stored and for each of them a respective Product class is created, as explained earlier in Section 3.2.

### 3.6. KNN algorithm to find Products

After selecting the Ids of Products that the user has not yet bought, we create a Product class for each one, as explained in Section 3.2. As previously mentioned, we include the text revisions in our recommendation system. Therefore, we create a new database that contains, for each category of products, the 200 most common words used in the revisions which have an overall rating of 4 and 5.

In order to process the text revision, we split them into unique words and remove any punctuation or characters. After that, we remove the stop words, which are words that appear frequently in any document and do not add meaning to a sentence [20]. We also apply Lemmatization, which is the process that obtains the "lemma" of a word by understanding the part of speech and the context of the word in the respective sentence [26]. We choose Lemmatization over Stemming due to the fact that Lemmatization produces better results [27]. Finally, we select only the 200 most common words and save them in the new database.

Although the Products relate to Neighbours, we also want to consider the opinion of all persons that have purchased the product. Therefore, the TopFiveWords attribute of Product, consists in the extraction of the 5 most common words used in all the revisions of this Product. The process for picking these words is similar to the one described in the previous paragraph.

The last step of our ontology-based recommendation system consists of applying the KNN algorithm in order to find the nearest Products to recommend to the user. For each Product, we create a vector of features with three columns (see Figure 4). The first column is a value that will be given the value of 1 if the Product Price is between User MinPrice and User MaxPrice, and 0 if not. The second column identifies the overall rating given by the respective Neighbour, which can be a value between 1 and 5. The third column is a number between 0 and 5. This attribute is 0 if no word of the TopFiveWords is among in the 200 most common words of the product category and 5 if all words are. Figure 4 shows an example of the vector of features for three Products.

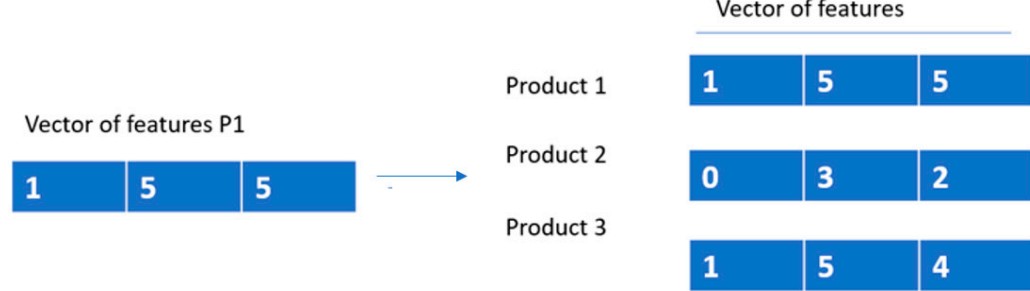

**Figure 4.** Creation of the vector of features to each Product.

We can see that Product 1 holds a Price that is between MinPrice and MaxPrice of the User's (value=1). Other than that, we can also see the respective Neighbour has given an overall rating of 5, which is the maximum value of an overall rating. The most 5 common words used in all the revisions of the Product are in the list of 200 most common words of the respective category given to the revisions with an overall rating of 4 and 5, which is also the maximum value. On the other hand, the Price of Product 2 is not in within range of MinPrice and MaxPrice of the User (value = 0); the respective neighbor gives a rating of 3, and only 2 words of the TopFiveWords are in the list of 200 most common words of the respective category given to the revisions with a rating of 4 and 5.

The final step, after normalizing all the columns with a value between 0 and 1, is calculating the Euclidean distance between the Products Vector of features and the Vector of features P1. The system will recommend the K Products with the lowest Euclidean distance values to the User.

## 4. Experimental Evaluation

In the experimental evaluation, we assess the performance of the proposed hybrid ontology-based recommendation system on two metrics: performance time to apply the KNN algorithm to Neighbours and then to Products, and the quality of the recommended products to the active user.

In order to perform the evaluation of our work, we compare the hybrid ontology-based recommendation system that we propose in this paper, to a collaborative filtering approach, which means that in the second one, we also apply the KNN algorithm to Neighbours and Products. To do this, we only consider the preferences of the active user to find his Neighbours and to find the Products we only consider the rating given by Neighbours.

For both versions of the recommendation system, we first compare the performance time of the applying KNN to find the nearest Neighbours. That is the time taken to create the vector of features to all the Neighbours and the time to apply the Euclidean distance between them and the User. Secondly, we compare the performance time of applying KNN to Products, which consists of the time taken to create the vector of features for all the Products and the time taken to apply the Euclidean distance to find the best Products to recommend to the active user.

We also intend to do a comparison between the number of nearest Neighbours and the number of nearest Products. To achieve this, we test the results when the number of nearest Neighbours is 1, 3 and 5 and when the number of nearest Products is 1, 3 and 5 which gives a total of 9 possibilities.

This section shows the results of the tests described above and their respective conclusions. In order to perform them, we randomly choose 4 users who have made revisions in our Reviews Database.

Figure 5 shows the profile of the four active users to which our hybrid ontology-based system will recommend new products. Figure 5 also shows the number of Neighbours of each active user and the time to apply KNN to find those Neighbours for both versions of the recommendation system. In order to help understand the experimental evaluation, the version of the recommendation system that concerns the hybrid ontology-based is called version A (see Figure 6), and version that concerns the collaborative filtering with KNN to find nearest Neighbours and Products is called version B (see Figure 7). In the next sections are detailed the results obtained in the experiments.

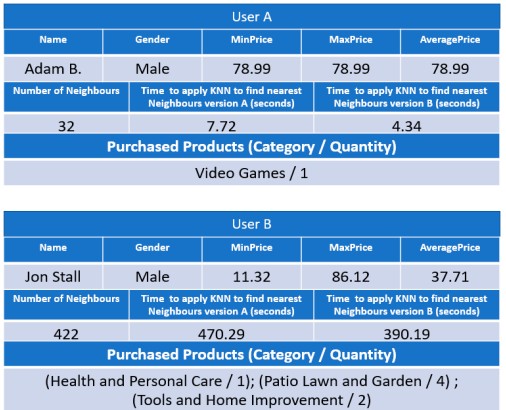

**Figure 5.** User Profiles to experimental evaluation.

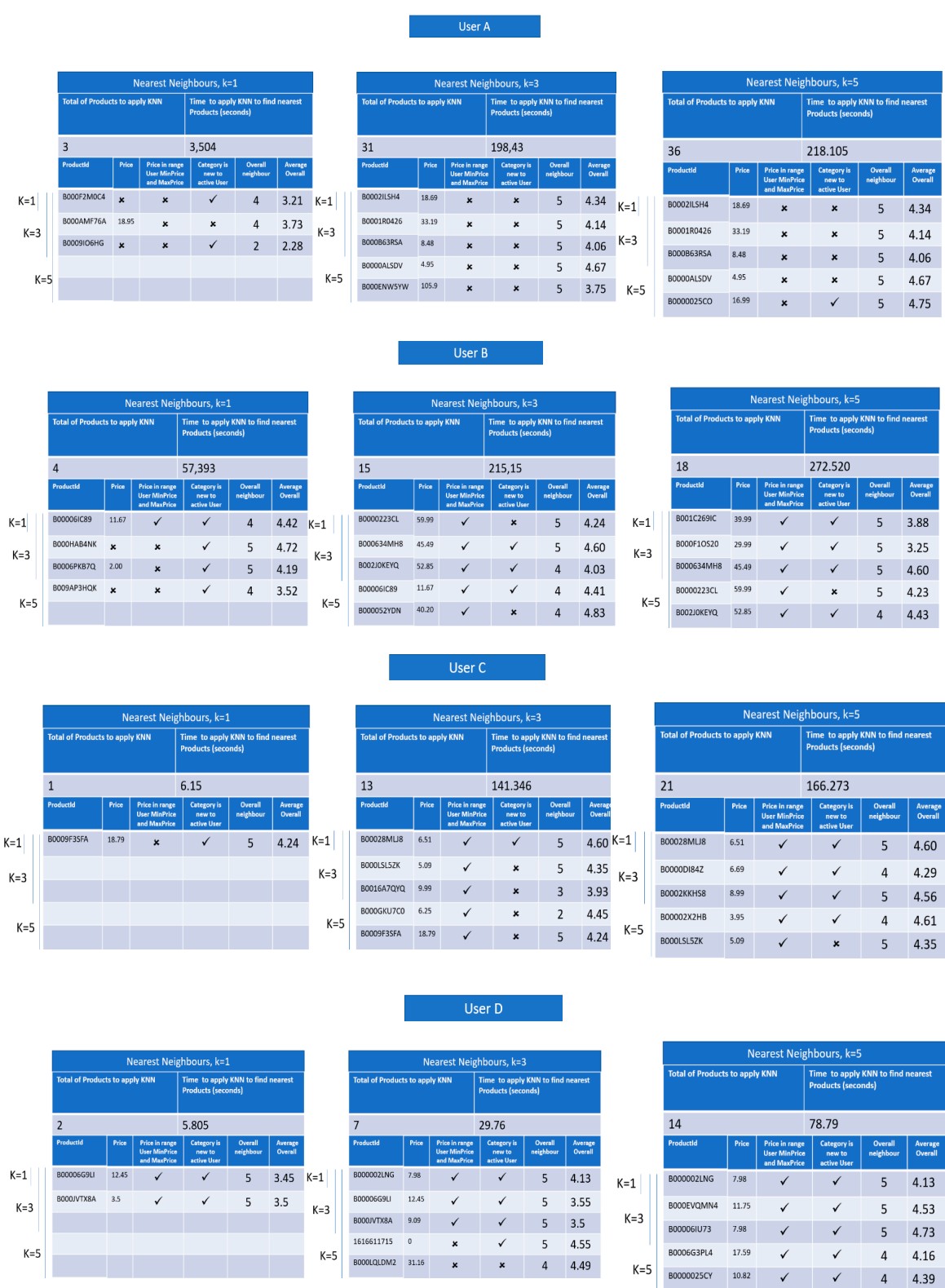

**Figure 6.** Version A of the recommendation system, and Products recommend to each active User.

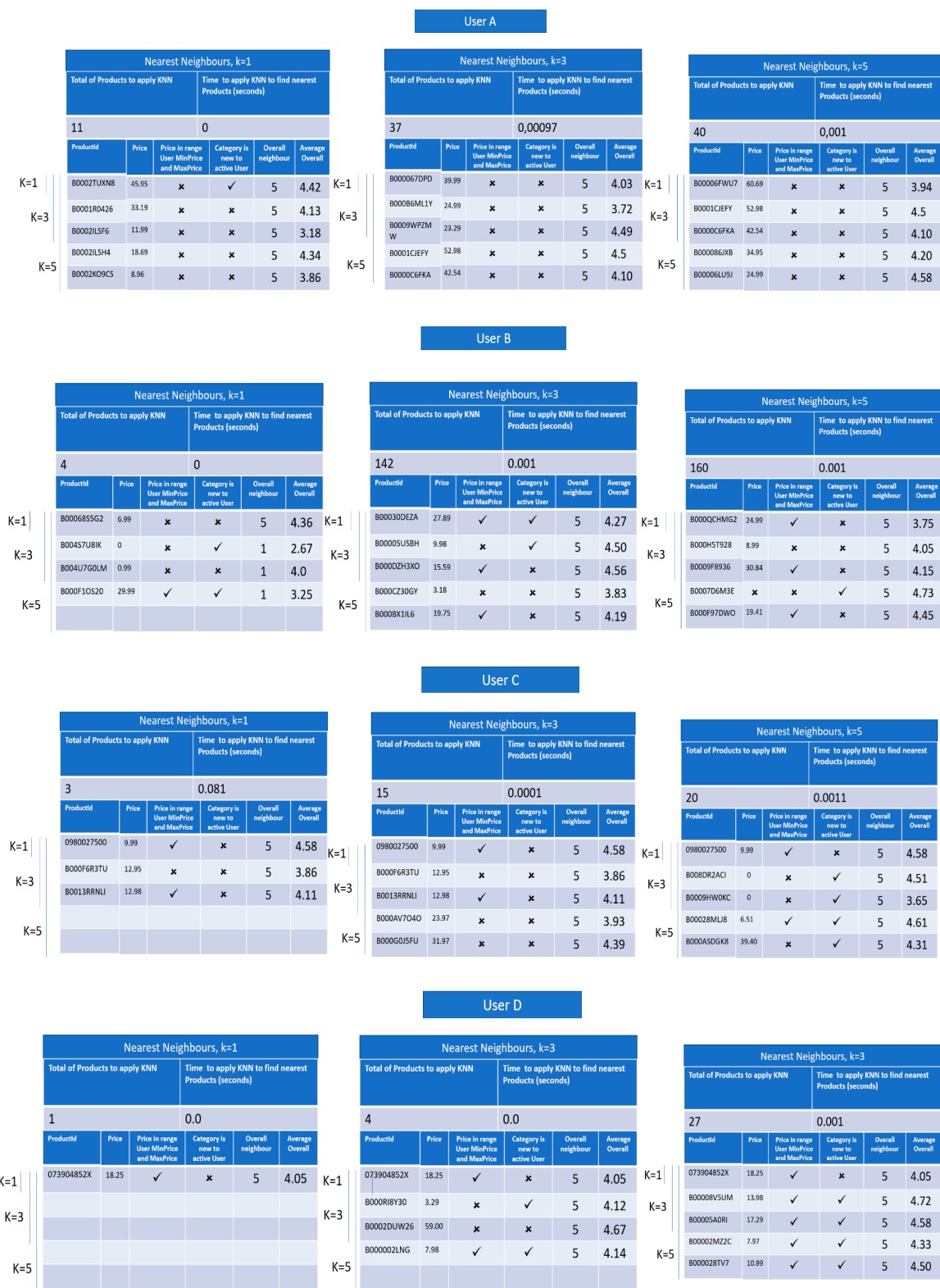

**Figure 7.** Version B of the recommendation system and Products recommend to each active User.

### 4.1. Time Performance of Applying KNN to Find Nearest Neighbours

Figure 5 presents the performance time of applying the KNN algorithm to find nearest Neighbours of both approaches, which consists of the time to create the vector of features and calculate the Euclidean Distance. For users A, C, and D, the time performance of applying KNN is identical for both versions with

a slight win towards version B, which achieves better performance time for users A and C. For user B, the time of applying the KNN algorithm is high for both versions, but version B also achieves better performance. These results can be explained by the time spent for creating the vector of features for each Neighbour and the number of Neighbours of each active user. Version A, which is the hybrid ontology-based recommender, also allows knowing the number of Products that each Neighbour has purchased in common with the active user in each category (only the categories that the active user has made reviews), it is also necessary to include the gender, the average price spent by the Neighbour in question and to find if each neighbour has purchased products in categories not purchased by the active user. We have also conclude that when the number of Neighbours increases the time of applying KNN also increases.

### 4.2. Time Performance of Applying KNN to Find Nearest Products

In this section, we compare the time performance of applying KNN to find the K-nearest Products to recommend to the active user. In our hybrid ontology-based recommender (version A, corresponds to Figure 6), a vector of features for each Product is created, which consists of three features, declared a Boolean value that indicates if the Product price is in range of MinPrice and MaxPrice of the active user; the rating given by the respective neighbor; and a feature that only concerns the Product: the TopFiveWords attribute, as explained in Section 3, for indicating in a range of 0 to 5, the number of words from the most 5 common words in all the reviews related to the product that are also in the 200 most common words used in reviews with overall rating of 4 and 5 in the respective product category. For the collaborative filtering recommender (version B, corresponds to Figure 7), we only use the ratings given by the respective Neighbour. Figures 6 and 7 show the results of the tests described above.

From the results presented in Figures 6 and 7, we observe that the time to find the k-nearest Products in version A is much higher than what is in version B. These results can be explained due to the time taken in extracingt the 5 most common words for each Product, which consists in extracting all the words used in all the reviews of the product, as explained before in Section 3.

Accordingly, we conclude that increasing the number of Products to which KNN will be applied, the time to apply it also increases. The time of apllying the KNN to the Products also increases depending on the number of reviews and words in each review of the Product. The lowest values of the time to apply the KNN in version B are explained by the simplicity of the vector of features, which only represents the overall rating given by the respective Neighbour.

### 4.3. Quality of the Recommended Products

Figures 6 and 7 also present a comparison between the Products recommended by the two versions of the recommendation system, the hybrid ontology-based (version A) and the collaborative filtering (version B). Based on the results shown in Figures 6 and 7, it is possible to confirm that our hybrid ontology-based recommendation system can not only recommend products from new categories to the active user, but can also match user cost preferences, even when we have an active user that has only purchased one product. This to say that means that the respective MinPrice and MaxPrice are equal. For user A, we see that in version B, the recommendation system only recommends one product that belongs to a new category, whilst the version A can recommend three.

In order to understand if the products recommended by the proposed approach are products that have received a high rating by all the users who have bought them, we extract the average overall rating given to each product that is recommended in the two versions of the recommendation system. Consequently, we conclude, by observing of the Average Overall column in Figures 6 and 7, that version A of the recommendation system can recommend new products to the active user that received a better overall rating when compared to the products recommended by version B.

In the experimental evaluation, we tested the combination of three values for the k-nearest Neighbours and the k-nearest Products. The results show that the combination, which achieved best results is when k is 5 in both KNN applications. In other words, when we choose to find the 5 nearest

Neighbours of the active user and recommend the 5 nearest Products is when the products recommended match user's preferences and belong to new categories to the active user.

From the experimental results, it is also possible to conclude that in version A of the recommendation system, the number of products which the KNN algorithm applies to find the nearest ones is lower when compared to version B. Consequently, the version A of the recommendation system can find Neighbours that are similar to the active user, which is translated into products that better satisfy the active user's preferences and demands. Contrariwise, in spite of version B of the recommendation system provides a higher number of Products to which the KNN algorithm will be applied to, those products don't easily match the user preferences and belong to categories to which the active user has already bought products from.

Figure 8 presents the importance of the TopFiveWords attribute of each Product to provide better recommendations. For each product recommended by the hybrid ontology-based, the figure shows the relation between the number of the 5 most common words that are in the 200 most common words used in the reviews with a rating of 4 or 5 in the respective category, and the average overall rating given by all the users that have bought the product.

| ProductId | TopFiveWords | Average Overall | | ProductId | TopFiveWords | Average Overall |
|---|---|---|---|---|---|---|
| B0009F3SFA | 5 | 4.05 | | B00006IC89 | 4 | 4.42 |
| B00028MLJ8 | 4 | 4.60 | | B000HAB4NK | 4 | 4.72 |
| B000LSL5ZK | 2 | 4.35 | | B0006PKB7Q | 3 | 4.19 |
| B0016A7QYQ | 3 | 3.93 | | B009AP3HQK | 3 | 3.52 |
| B000GKU7C0 | 2 | 4.45 | | B0000223CL | 4 | 4.24 |
| B0000DI84Z | 4 | 4.29 | | B000634MH8 | 4 | 4.60 |
| B0002KKHS8 | 3 | 4.56 | | B002J0KEYQ | 5 | 4.43 |
| B00002X2HB | 3 | 4.61 | | B000052YDN | 4 | 4.83 |
| B000LSL5ZK | 2 | 4.35 | | B001C269IC | 5 | 3.88 |
| B000F2M0C4 | 4 | 3.21 | | B00006G9LI | 3 | 3.45 |
| B000AMF76A | 4 | 3.73 | | B000JVTX8A | 2 | 3.5 |
| B0009IO6HG | 4 | 2.28 | | B000002LNG | 4 | 4.13 |
| B0002ILSH4 | 5 | 4.34 | | B00006G9LI | 3 | 3.55 |
| B0001R0426 | 5 | 4.14 | | 1616611715 | 5 | 4.55 |
| B000B63RSA | 5 | 4.06 | | B000LQLDM2 | 3 | 4.49 |
| B0000ALSDV | 5 | 4.67 | | B000EVQMN4 | 4 | 4.53 |
| B000ENW5YW | 4 | 3.75 | | B00006IU73 | 4 | 4.73 |
| B0000025CO | 4 | 4.75 | | B0006G3PL4 | 5 | 4.16 |
| | | | | B0000025CY | 4 | 4.39 |

**Figure 8.** Relation between TopFiveWords and average overall of each Product.

From Figure 8, we can affirm that most of the times the TopFiveWords attribute it is similar to the average overall rating given to that product. Therefore, when the average overall rating of the Product increases the TopFiveWords also increases and the same for when the average overall rating goes down.

## 4.4. Discussion of the Results

From the the experimental evaluation, we can conclude that the hybrid ontology-based recommender, proposed in this paper, improves the recommendation of Products in e-commerce. Mainly, it increases the number of Products recommended that belong to unknown categories to the

active User. The Products that are recommended to the active User also match his preferences more easily when compared to the collaborative filtering version of the recommendation system. Even when the active User has only bought one Product, which causes some missing features of the active User, such as the MinPrice and Max Price being equal.

However, our hybrid ontology-based recommender consumes more time to apply KNN to find the k-nearest Products. As explained previously, this is strongly related to the time consuming for extracting the 5 most common words used in all the reviews of that product, which consists of extracting the words, removing stop words, applying Lemmatization and then finding the 5 most common. In inverse, the time to apply KNN to find the k-nearest Products in collaborative filtering is very low, because it only focuses on choosing the Products with the highest ratings given by the respective Neighbours.

## 5. Conclusions and Future Work

We proposed a new approach to recommend Products in e-commerce systems, which combines the simplicity of finding users that have similar preferences with the active user, with ontology-based models. This approach generates more knowledge about the active User, the respective Neighbours and the Products recommended to them (the user) and the relation between these three.

The experimental evaluation shows that our hybrid ontology-based system can recommend, in comparison with the collaborative filtering version, products that match user preferences more easily and belong to new categories for the active user. The proposed approach also chooses the k-Nearest Neighbours of the active user, which is translated in a small number of products to which KNN will be applied to. However, it satisfies the active user's preferences, when compared to the collaborative filtering version, which provides a large number of products to which KNN will be applied, but most of the time, belong to categories, which that active user already knows.

The experimental evaluation also shows that the hybrid ontology-based recommender, in comparison with the collaborative filtering version, can recommend products, which have received a higher average overall rating, to the active user.

Despite the good results achieved by the hybrid ontology-based recommendation system, the time to apply the KNN to find the k-nearest Products is too high. For this reason, as in future work, in order to reduce the time of applying the KNN algorithm to find the k-nearest Products, we intend to find the 5 most common words used in the all the reviews of each product and save this attribute in the database, which will dramatically reduce the time of applying the KNN algorithm to the Products. Although generally, the system can discover the active users's gender based on the reviewer name, it would be interesting to find a way to discover his/her gender when the reviewer's name is a nickname or is based on numbers. This procedure might increase the quality of the recommended products.

As future work, we also intend to apply the proposed approach to the entire dataset, which as previously mentioned, consists of 142.8 million reviews. In order to apply the KNN algorithm to find the k-nearest products, we also plan to take into consideration of the vector of features' creation and the average overall rating of each product.

We also intend to improve our models by adding some extra features. In this paper, we have worked, with English texts, then we extend this to other language texts. We will also try to detect other sentiment labels of the human being and at the same time, working with a big amount of texts. Finally, it would be of interest to work with other datasets and perform evaluation based on more measures such as accuracy, precision, recall, MAE (Mean Absolute Error), and also NDCG (Normalized Discounted Cumulative Gain).

**Author Contributions:** Conceptualization, R.R.S. and J.B.; Methodology, M.G. and R.R.S.; Software, M.G.; Validation, M.G., R.R.S. and J.B.; Formal analysis, M.G., R.R.S. and J.B.; Investigation, M.G.; Resources, M.G.; Data curation, M.G.; Writing—original draft preparation, M.G.; Writing—review and editing, M.G., R.R.S. and J.B.; Supervision, R.R.S. and J.B.; Project administration, R.R.S. and J.B.; Funding acquisition, J.B.

**Funding:** This research received no external funding.

**Acknowledgments:** We acknowledge Centre of Informatics and Systems of University of Coimbra (CISUC) for the facilities offered during this project.

**Conflicts of Interest:** The authors declare no conflict of interest.

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
