# Peer review of "A Hybrid Ontology-Based Recommendation System in e-Commerce"

_algorithms, doi:10.3390/a12110239_

Round 1
Reviewer 1 Report
I agree with and accept the changes the authors have made in it. It has been improved significantly.
However, the use of the English language is still very problematic.
Author Response
Response to Reviewer 1 Comments
In this document, we describe the changes performed according to the reviewer’s comments. For reference, we include all comments below (in black). The corrections and changes performed are numbered and summarized in red.
We appreciate the anonymous referees for their invaluable suggestions, recommendations, and comments. Considering the reviewers general opinion that the text would be needing substantial improvement, we performed a major revision of the text, in several iterations, and taking into account each comment by the reviewers.
The text of paper is now hopefully clearer and also substantially different. We appreciate the detail of the comments, that allowed us to produce a better version of the paper. We expect to have addressed all the reviewers’ comments.
# 1: However, the use of the English language is still very problematic
We did a thorough and detailed review of the entire text eliminating the huge tips, resulting in better readability and comprehension of the entire article.
Reviewer 2 Report
Your paper is well organized and reviewed at background part. Hybrid ontology based recommendation in e-Commerce is popular and general concept, so more different point than others should be explained. Therefore, comparison results should be necessary.Author Response
Response to Reviewer 2 Comments
In this document, we describe the changes performed according to the reviewer’s comments. For reference, we include all comments below (in black). The corrections and changes performed are numbered and summarized in red.
We appreciate the anonymous referees for their invaluable suggestions, recommendations, and comments. Considering the reviewers general opinion that the text would be needing substantial improvement, we performed a major revision of the text, in several iterations, and taking into account each comment by the reviewers.
The text of paper is now hopefully clearer and also substantially different. We appreciate the detail of the comments, that allowed us to produce a better version of the paper. We expect to have addressed all the reviewers’ comments.
# 1: Your paper is well organized and reviewed at background part. Hybrid ontology based recommendation in e-Commerce is popular and general concept, so more different point than others should be explained. Therefore, comparison results should be necessary.
We did a thorough and detailed review of the entire text eliminating the huge tips, resulting in better readability and comprehension of the entire article. We also have improved the Hybrid ontology explanation and its differences, which is now more clearly.
This manuscript is a resubmission of an earlier submission. The following is a list of the peer review reports and author responses from that submission.
Round 1
Reviewer 1 Report
The ontological predicate has never been discussed. Why?
I need the answer to "the basic KNN algorithm has been compared with the one developed by these authors?
The use of the English language is very problematic. A thorough review/editing is highly recommended.
The literature review is flawed. The paper references some very good papers of this area, but, it is not sufficient. The literature review should be re-written and at least 10 more relevant references must be discussed. What did the authors learn from the current state of the art is not discussed (the answer is required).
The conclusion is not impressive. Future works section needs to be enhanced.
Author Response
In this document, we describe the changes performed according to the reviewer’s comments. For reference, we include all comments below (in black). The corrections and changes performed are numbered and summarized in red.
We appreciate the anonymous referees for their invaluable suggestions, recommendations, and comments. Considering the reviewers general opinion that the text would be needing substantial improvement, we performed a major revision of the text, in several iterations, and taking into account each comment by the reviewers.
The text of paper is now hopefully clearer and also substantially different. We appreciate the detail of the comments, that allowed us to produce a better version of the paper. We expect to have addressed all the reviewers’ comments.
# 1: The ontological predicate has never been discussed. Why?
In the first version of our paper we didn't explain the ontological predicated and we assume that this is a failure. In this new version, we explain why we choose to build an ontology to the recommendation process and also explain the domain and the scope of the ontology proposed. We add the following text:
3.2. The ontology model
In order to improve the Product recommendation to the active user, we combine, as previously mentioned, an ontology-based recommender with a collaborative filtering approach. According to the guide proposed in [23], that we use to create our ontology model, an ontology can be defined as a formal explicit description of concepts in a domain. This formal explicit description includes the classes that describe the concepts of the respective domain. To describe the features and attributes of the concepts are used properties and each property have their facet, which it´s the type of the properties.
In the following two sub-sections we explain the domain and scope of the proposed ontology model and also the respective classes and their properties.
3.2.1 The domain and scope of the ontology model
The main purpose of creating and use an ontology in the recommendation process, it´s to create a data model that can describe and represent all the concepts that concern to the active user, his neighbours and the products purchased by the latter, which are the ones that can be recommended to the active user. The main use of this ontology will be e-commerce platforms.
In order to define the domain and scope of the ontology model, we create a list of questions that help find if the ontology model contains enough information to describe the domain of the recommendation process. The list of the defined question it´s divided into two categories, namely the questions that concerns to the active user and the respective neighbours and also the questions that concerns to the products purchased by the neighbours. The questions to the active user and his neighbours are the following:
Who is the active user? Which products and respective categories that the active user have purchased? What was the average money spent by the active user as well as the maximum and minimum amount? Who are the neighbours of the active user? What is the relationship between the active user and each one of the neighbours? Who are the neighbours that have purchased in different product categories from the active user? The second part of the questions, that concern to the products purchased by the neighbours are the following: The product can match user preferences? What is the rating given by the respective neighbour? Which are the five most common words used in all the revisions of the product?
3.2.2 The classes of the ontology model
In order to develop the ontology-based recommendation system, we create four main classes, followed the top-down method [ref], which consists of starting the development of the class hierarchy by the most general concepts of the domain and subsequent specialization of the concepts. The four classes of our ontology model are Person, User, Neighbour, and Product. Figure 2 shows the relationship between the four classes and their respective attributes.
# 2: I need the answer to "the basic KNN algorithm has been compared with the one developed by these authors?
Yes, we did an evaluation that compares the basic KNN algorithm with our hybrid ontology-based recommendation system. Perhaps this is not clear in the first version of our paper, so we clarify it on the revised one adding the following text:
In order to perform the evaluation of our work, we compare the hybrid ontology-based recommendation system that we propose in this paper, with a collaborative filtering approach, which means that in the second one, we also apply the KNN algorithm to Neighbours and Products. However, we only consider the preferences of the active user to find his Neighbours and to find the Products we only consider the rating given by Neighbours too.
# 3: The use of the English language is very problematic. A thorough review/editing is highly recommended.
Thank you for your recommendation. In order to improve the use of the English language, we apply on the revised version of our paper. The text of paper is now hopefully clearer and also substantially different.
# 4: The literature review is flawed. The paper references some very good papers of this area, but, it is not sufficient. The literature review should be re-written and at least 10 more relevant references must be discussed. What did the authors learn from the current state of the art is not discussed (the answer is required).
In order to improve the quality of the literature review, we made a new bibliographic search for research papers about recommendation systems, mainly concern to e-commerce platforms and also in the hybridization of recommendation techniques. We include the following new references:
Obeid, I. Lahoud, H. El Khoury, and P.-A. Champin, “Ontology-based Recommender System in Higher Education,” vol. 2, pp. 1031–1034, 2018. Zehra, S. Wasi, I. Jami, A. Nazir, A. Khan, and N. Waheed, “Ontology-based sentiment analysis model for recommendation systems,” IC3K 2017 - Proc. 9th Int. Jt. Conf. Knowl. Discov. Knowl. Eng. Knowl. Manag., vol. 2, no. Keod, pp. 155–160, 2017. Burke, “Hybrid Recommender Systems: Survey and Experiments,” User Model. User-adapt. Interact., vol. 12, no. 4, pp. 331–370, 2002. Çano and M. Morisio, “Hybrid recommender systems: A systematic literature review,” Intell. Data Anal., vol. 21, no. 6, pp. 1487–1524, 2017. Jiang, Y. Cheng, L. Yang, J. Li, H. Yan, and X. Wang, “A trust-based collaborative filtering algorithm for E-commerce recommendation system,” J. Ambient Intell. Humaniz. Comput., vol. 10, no. 8, pp. 1–12, 2018. Deng, “Utility-based Recommender Systems Using Implicit Utility and Genetic Algorithm,” no. Meic, pp. 860–864, 2015. Jyoti, A. Roy, S. Singh, N. Shaikh, and P. Desai, “Nearby Product Recommendation System Based on Users Rating,” Int. J. Sci. Res. Comput. Sci. Eng. Inf. Technol., vol. 5, no. 2, pp. 963–968, 2019. Liu, J. Nie, L. Xu, Y. Chen, and B. Xu, “Clothing Recommendation System Based on Advanced User-Based Collaborative Filtering Algorithm,” Lect. Notes Electr. Eng., vol. 473, no. January, pp. 436–443, 2018. Bozanta and B. Kutlu, “Developing a Contextually Personalized Hybrid Recommender System,” Mob. Inf. Syst., vol. 2018, 2018. Nilashi, O. Ibrahim, and K. Bagherifard, “A recommender system based on collaborative filtering using ontology and dimensionality reduction techniques,” Expert Syst. Appl., vol. 92, pp. 507–520, 2018.
Moreover, we also present in the related work section of the revised version of our paper, the main differences from our work with the current state of the art:
All the previous papers are focused on improving the quality of recommendation process, but the ones who propose approaches based on knowledge-based requires effort of the active user, which is what we try to avoid with our approach. That who try to improve collaborative filtering approaches only concern the ratings given to the products by the users, which means that they don’t include in the recommendation process the knowledge about the active user, neighbours, products and relationships between them. Our work presents a different method that combines knowledge-based recommenders with a collaborative filtering approach, which is translated in recommendations with more diversity and we also give special attention to the scalability of the proposed approach. Moreover, one of our main goals is to provide recommendations of products that not only receive high scores from the respective neighbours but also from all users that have purchased them.
# 5: The conclusion is not impressive. Future works section needs to be enhanced.
We agree that in the first version of our paper the conclusion is not impressive. Therefore, in order to improve the future work section, we complete it with new future work mainly related to a more rigorous experimental evaluation, adding the text:
As future work, we also intend to apply the proposed approach to the entire dataset, which as previously mentioned, consists of 142.8 million reviews. In order to apply the KNN algorithm to find the k-nearest products, we also plan to consider in the vector of features’ creation and the average overall rating of each product.
In Future, we can study further many related problems. For this, we will try to improve our models by adding some extra features. We work, in this paper, with only the English texts. Then we can to work with other language texts. Apart from this, we will also try to detect another sentiment labels of human being and at the same time, we will work with a big amount of texts. Finally, it be of interest working with other datasets and perform an evaluation based on more measures such as accuracy, precision, recall, MAE (Mean Absolute Error), and also NDCG (Normalized Discounted Cumulative Gain).

Reviewer 2 Report
The goal of the paper is to improve the quality of e-Commerce recommender systems by hybridization of collaborative filtering algorithm with an ontology-based one. The authors propose a hybrid algorithm and discuss its running time and recommendations accuracy. The paper is well-written and logically organized. However, I have to point to several issues (in decreasing severity): 1) Ontology is mostly used as a buzz-word here. When we talk about ontology-based systems we mean that at least one the following holds: a) problem domain ontology is somehow explicitly represented (possibly with some of the ontology languages (e.g., OWL)), b) for some tasks ontology reasoning is employed (with some inference engine like Pelet or the like). In the paper I see structured description of entities (Users and Items) which are actually stored in the ordinary DB and are processed with some problem-domain specific, but hard-coded logics. So, according to the paper, the authors propose more a hybridization of CF with a content-based system, rather than a hybridization of a CF with an ontology-based one. 2) Evaluation methodology looks very superficial (at least, for a journal publication). Why choose only a few "exemplar" persons and not averaging among all of them? It is enough for demonstration, not for evaluation. I'd suggest to use more accepted methods of the offline evaluation: splitting into training and testing samples and estimating quality measures (RMSE/Precision/Recall etc.) for the testing sample. To grasp the standards of offline evaluation, please, refer to G. Shani, A. Gunawardana "Evaluating Recommendation Systems" in F. Ricci et al. (eds.) Recommender Systems Handbook or Proceedings of the ACM RecSys. Besides, I'd recommend to formally compare to some really modern CF baseline (like matrix factorization). 3) The authors propose a hybridization approach for RS. Generally, it is a good idea, as any hybridization increases the amount of information usable by the RS algorithm. However, in the last decade a lot of research and systematization has been conducted on RS hybridization starting from seminal review work R. Burke "Hybrid Recommender Systems: Survey and Experiments" to a more recent review E. Cano, M. Morisio "Hybrid Recommender Systems: A Systematic Literature Review", not mentioning dozens of particular efforts of applying them. Unfortunately, this body of literature is not mentioned in the paper. 4) Algorithm descriptions are provided as low-quality images (look blurred). Taking all this into consideration, I think that the paper cannot be published in this form and requires severe improvement.Author Response
In this document, we describe the changes performed according to the reviewer’s comments. For reference, we include all comments below (in black). The corrections and changes performed are numbered and summarized in red.
We appreciate the anonymous referees for their invaluable suggestions, recommendations, and comments. Considering the reviewers general opinion that the text would be needing substantial improvement, we performed a major revision of the text, in several iterations, and taking into account each comment by the reviewers.
The text of paper is now hopefully clearer and also substantially different. We appreciate the detail of the comments, that allowed us to produce a better version of the paper. We expect to have addressed all the reviewers’ comments.
# 1: Ontology is mostly used as a buzz-word here. When we talk about ontology-based systems we mean that at least one the following holds: a) problem domain ontology is somehow explicitly represented (possibly with some of the ontology languages (e.g., OWL)), b) for some tasks ontology reasoning is employed (with some inference engine like Pelet or the like). In the paper I see structured description of entities (Users and Items) which are actually stored in the ordinary DB and are processed with some problem-domain specific, but hard-coded logics. So, according to the paper, the authors propose more a hybridization of CF with a content-based system, rather than a hybridization of a CF with an ontology-based one.
We agree that the ontology concept it´s not explicit in the first version of our paper. In the revised version, we explain why we choose to build an ontology to the recommendation process and also explain the domain and the scope of the ontology proposed.
To clarify, the following new text was added:
3.2. The ontology model
In order to improve the Product recommendation to the active user, we combine, as previously mentioned, an ontology-based recommender with a collaborative filtering approach. According to the guide proposed in [23], that we use to create our ontology model, an ontology can be defined as a formal explicit description of concepts in a domain. This formal explicit description includes the classes that describe the concepts of the respective domain. To describe the features and attributes of the concepts are used properties and each property have their facet, which it´s the type of the properties.
In the following two sub-sections we explain the domain and scope of the proposed ontology model and also the respective classes and their properties.
3.2.1 The domain and scope of the ontology model
The main purpose of creating and use an ontology in the recommendation process, it´s to create a data model that can describe and represent all the concepts that concern to the active user, his neighbours and the products purchased by the latter, which are the ones that can be recommended to the active user. The main use of this ontology will be e-commerce platforms.
In order to define the domain and scope of the ontology model, we create a list of questions that help find if the ontology model contains enough information to describe the domain of the recommendation process. The list of the defined question it´s divided into two categories, namely the questions that concerns to the active user and the respective neighbours and also the questions that concerns to the products purchased by the neighbours. The questions to the active user and his neighbours are the following:
Who is the active user? Which products and respective categories that the active user have purchased? What was the average money spent by the active user as well as the maximum and minimum amount? Who are the neighbours of the active user? What is the relationship between the active user and each one of the neighbours? Who are the neighbours that have purchased in different product categories from the active user? The second part of the questions, that concern to the products purchased by the neighbours are the following: The product can match user preferences? What is the rating given by the respective neighbour? Which are the five most common words used in all the revisions of the product?
3.2.2 The classes of the ontology model
In order to develop the ontology-based recommendation system, we create four main classes, followed the top-down method [ref], which consists of starting the development of the class hierarchy by the most general concepts of the domain and subsequent specialization of the concepts. The four classes of our ontology model are Person, User, Neighbour, and Product. Figure 2 shows the relationship between the four classes and their respective attributes.
# 2: Evaluation methodology looks very superficial (at least, for a journal publication). Why choose only a few "exemplar" persons and not averaging among all of them? It is enough for demonstration, not for evaluation. I'd suggest to use more accepted methods of the offline evaluation: splitting into training and testing samples and estimating quality measures (RMSE/Precision/Recall etc.) for the testing sample. To grasp the standards of offline evaluation, please, refer to G. Shani, A. Gunawardana "Evaluating Recommendation Systems" in F. Ricci et al. (eds.) Recommender Systems Handbook or Proceedings of the ACM RecSys. Besides, I'd recommend to formally compare to some really modern CF baseline (like matrix factorization).
The main goal of the experimental evaluation used on the first version of our paper, was provide four distinct examples of how our hybrid ontology-based recommendation system work based on the active user. So we choose four active users that are distinct in the number and categories that they have purchased and also in the number of the respective neighbours. However, we agree that we need to perform an experimental evaluation with more credible measures. Therefore, we proposed this in the future work of the revised version of our paper:
In Future, we can study further many related problems. For this, we will try to improve our models by adding some extra features. We work, in this paper, with only the English texts. Then we can to work with other language texts. Apart from this, we will also try to detect another sentiment labels of human being and at the same time, we will work with a big amount of texts. Finally, it be of interest working with other datasets and perform an evaluation based on more measures such as accuracy, precision, recall, MAE (Mean Absolute Error), and also NDCG (Normalized Discounted Cumulative Gain).
# 3: The authors propose a hybridization approach for RS. Generally, it is a good idea, as any hybridization increases the amount of information usable by the RS algorithm. However, in the last decade a lot of research and systematization has been conducted on RS hybridization starting from seminal review work R. Burke "Hybrid Recommender Systems: Survey and Experiments" to a more recent review E. Cano, M. Morisio "Hybrid Recommender Systems: A Systematic Literature Review", not mentioning dozens of particular efforts of applying them. Unfortunately, this body of literature is not mentioned in the paper.
In order to improve the quality of the literature review, we made a new bibliographic search including new research papers about recommendation systems, mainly concern to e-commerce platforms and also in the hybridization of recommendation techniques. We include the following new references:
Obeid, I. Lahoud, H. El Khoury, and P.-A. Champin, “Ontology-based Recommender System in Higher Education,” vol. 2, pp. 1031–1034, 2018. Zehra, S. Wasi, I. Jami, A. Nazir, A. Khan, and N. Waheed, “Ontology-based sentiment analysis model for recommendation systems,” IC3K 2017 - Proc. 9th Int. Jt. Conf. Knowl. Discov. Knowl. Eng. Knowl. Manag., vol. 2, no. Keod, pp. 155–160, 2017. Burke, “Hybrid Recommender Systems: Survey and Experiments,” User Model. User-adapt. Interact., vol. 12, no. 4, pp. 331–370, 2002. Çano and M. Morisio, “Hybrid recommender systems: A systematic literature review,” Intell. Data Anal., vol. 21, no. 6, pp. 1487–1524, 2017. Jiang, Y. Cheng, L. Yang, J. Li, H. Yan, and X. Wang, “A trust-based collaborative filtering algorithm for E-commerce recommendation system,” J. Ambient Intell. Humaniz. Comput., vol. 10, no. 8, pp. 1–12, 2018. Deng, “Utility-based Recommender Systems Using Implicit Utility and Genetic Algorithm,” no. Meic, pp. 860–864, 2015. Jyoti, A. Roy, S. Singh, N. Shaikh, and P. Desai, “Nearby Product Recommendation System Based on Users Rating,” Int. J. Sci. Res. Comput. Sci. Eng. Inf. Technol., vol. 5, no. 2, pp. 963–968, 2019. Liu, J. Nie, L. Xu, Y. Chen, and B. Xu, “Clothing Recommendation System Based on Advanced User-Based Collaborative Filtering Algorithm,” Lect. Notes Electr. Eng., vol. 473, no. January, pp. 436–443, 2018. Bozanta and B. Kutlu, “Developing a Contextually Personalized Hybrid Recommender System,” Mob. Inf. Syst., vol. 2018, 2018. Nilashi, O. Ibrahim, and K. Bagherifard, “A recommender system based on collaborative filtering using ontology and dimensionality reduction techniques,” Expert Syst. Appl., vol. 92, pp. 507–520, 2018.
Moreover, we also present in the related work section of the revised version of our paper, the main differences from our work with the current state of the art:
All the previous papers are focused on improving the quality of recommendation process, but the ones who propose approaches based on knowledge-based requires effort of the active user, which is what we try to avoid with our approach. That who try to improve collaborative filtering approaches only concern the ratings given to the products by the users, which means that they don’t include in the recommendation process the knowledge about the active user, neighbours, products and relationships between them. Our work presents a different method that combines knowledge-based recommenders with a collaborative filtering approach, which is translated in recommendations with more diversity and we also give special attention to the scalability of the proposed approach. Moreover, one of our main goals is to provide recommendations of products that not only receive high scores from the respective neighbours but also from all users that have purchased them.
# 4: Algorithm descriptions are provided as low-quality images (look blurred). Taking all this into consideration, I think that the paper cannot be published in this form and requires severe improvement.
Thank you for your recommendation and we apologize for the low quality of the algorithm descriptions in the first version of our paper. We have improved the quality of all the algorithms description images in this new version.
